

# Converging Profile Relationships for Offshore Wind Speed and Turbulence Intensity

Gus Jeans [1]

[1] Oceanalysis Ltd, Wallingford, OX10 0DD, UK

*Correspondence to*: Gus Jeans (gus.jeans@oceanalysis.com)

**Abstract.** This paper reduces uncertainty in the quantification of offshore wind speed and turbulence intensity. A primary application is estimation of extreme wind speeds for design of wind energy generation systems, including turbines and fixed or floating support structures. Results of significance to normal winds, for resource assessment, engineering design or operations, can also be derived. This research is part of wider efforts to bring together the long established, but traditionally

separate, onshore wind energy and metocean technical disciplines. A range of industry standard relationships, including those from International Electrotechnical Commission (IEC) and International Organization for Standardization (ISO), are compared with an extensive set of met mast data collected offshore Northwestern Europe over recent decades. Analysis initially focused on over 1000 independent storm peak events identified within the 10 minute mean wind speed time series. Time series and coherent vertical profiles were subjected to detailed scrutiny and analysis, considering wind speeds at

various averaging periods, turbulence intensity and gust factors. Most peak events were associated with neutral atmospheric conditions, so were well represented by the ISO Frøya profile, with shear close to the IEC power law exponent of $\alpha = 0.11$. A new pragmatic framework for classification of wind profiles in terms of relative shear is outlined, bringing together key elements of the IEC and ISO standards. This relative shear framework, which extends over the full range of measured wind speeds, is related to atmospheric stability and used to distinguish different classes of ambient turbulence intensity. New

empirical relationships to quantify offshore wind ambient turbulence intensity are described. This study highlights the critical role of turbulence intensity in the estimation of gust factors, with a range of relevant relationships from IEC, ISO and other sources assessed using the extensive set of measured storm peak events. A simple generalised form of gust factor relationship is adopted, with a coefficient that varies with averaging period. A similar, but distinct, analysis yields recommendations for estimating peak 10 minute mean winds from peaks in 1 or 3 hour mean winds. Finally, a simplified

workflow for the estimation of extreme offshore winds is outlined.

**Short summary.** An extensive set of met mast data offshore Northwestern Europe is used to reduce uncertainty in offshore wind speed and turbulence intensity. The performance of widely used industry standard relationships is quantified, while some new empirical relationships are derived for practical application. Motivations include encouraging appropriate

convergence of traditionally separate technical disciplines within the rapidly growing offshore wind energy industry.



## 1 Introduction

The wind energy industry evolved onshore, where many of the standard relationships and established practices are still focussed. These are established within the International Electrotechnical Commission (IEC) standards. The quantification of offshore winds for engineering design and operations has a long history in oil and gas, leading to relationships established by International Organization for Standardization (ISO). The development of offshore turbine guidelines by IEC makes extensive reference to relevant ISO standards, but some differences persist between the two traditions. These can lead to inconsistencies and potentially waste time and effort during wind energy projects. The motivations for this study include efforts to help resolve some of these inconsistencies and encourage appropriate convergence of the technical disciplines. This is now especially important as the energy transition accelerates, with an increasing number of established offshore energy operators rapidly diversifying into wind power.

This paper considers three types of vertical profile relationship, which are not independent but closely related to each other. The first type of relationship considers profiles of mean wind speed. At any point above the sea surface, there are an infinite number of possible mean wind speed values, each corresponding to a different averaging period. In practical terms, averaging periods from a few seconds to several decades are of interest for offshore wind energy. Most engineering applications require wind speed averages over periods of 10 minutes or 1 hour. Shorter duration gusts are considered below, very appropriately after turbulence intensity.

In broad terms, the IEC guidelines specify power laws while ISO tends to prefer logarithmic profiles. Both can provide a pragmatic representation of the wind speed profile in the surface layer in the absence of strong atmospheric stability or instability. Excellent reviews of relevant theory are provided by texts such as Emeis (2018) and Landberg (2015), so are not duplicated here. A power law gives the wind speed $U(z)$ at height $z$ as a function of wind speed $U(z_r)$ at height $z_r$ according to Eq. (1):

$$U(z) = U(z_r) \left(\frac{z}{z_r}\right)^{\alpha} \tag{1}$$

The power law exponent $\alpha$ is widely known as the shear parameter in wind energy applications. A value of $\alpha = 0.11$ is given for 50 year extreme profiles of 10 minute mean wind speed for onshore wind turbines by IEC 61400-1:2019. The same value is inherited by IEC 61400-3-1:2019 for offshore turbines, which also gives $\alpha = 0.14$ for normal offshore wind conditions. The following Frøya logarithmic profile is given by ISO 19901-1:2015 for offshore storm conditions, characterised by nearly neutral atmospheric stability:

$$U(z) = U(z_r) \left[1 + C \cdot \ln\left(\frac{z}{z_r}\right)\right] \tag{2}$$



where $C = 0.0573 \cdot \sqrt{1 + 0.15\ U(z_r)}$ and $U(z_r)$ is the 1 hour mean wind speed at $z_r = 10$ m above the sea surface. The ISO Frøya profile is a special case of the logarithmic profile, in which the roughness length can be determined from the wind

speed at just one elevation. Pragmatically, $U(z_r)$ can be found by fitting any measured wind profile to Eq. (2). This paper will demonstrate that Eq. (2) can also be applied when $U(z_r)$ is the 10 minute mean wind speed at $z_r = 10$ m, of more direct relevance to offshore wind energy. This follows the fact that and $U(z_r)$ is the 40 minute mean wind speed at $z_r = 10$ m in the original Frøya formulation of Andersen and Løvseth (2006). The performance of the IEC and ISO relationships are compared in Sect. 3, then brought together in a new wind shear characterisation framework in Sect. 4.

The second type of vertical profile relationship considered in this paper is turbulence intensity. This is a critical parameter for wind turbine design that is often associated with considerable uncertainty in offshore wind energy projects, especially when site specific measurements are only available from floating LiDAR. Turbulence intensity plays a secondary role in some other long established wind engineering applications. It is often only an intermediate calculation used to determine appropriate gust factors, considered further below.

A range of turbulence intensity relationships from IEC 61400-1:2019, IEC 61400-3:2009 and IEC 61400-3-1:2019 will be briefly considered in Sect. 5 this paper, but the equations are not duplicated here. These include the offshore turbulence model outlined by Wang et al. (2014), referred to by IEC 61400-3-1:2019. Of more importance to this paper is the Frøya relationship for turbulence intensity $I_u(z)$ at height $z$ given by ISO 19901-1:2015:

$$I_u(z) = 0.06 \cdot [1 + 0.043\ U(z_r)] \left(\frac{z}{z_r}\right)^{-0.22} \tag{3}$$

where again $U(z_r)$ is the 1 hour mean wind speed at $z_r = 10$ m in the original formulation. This paper will suggest that taking $U(z_r)$ as the more directly available 10 minute mean wind speed adds a little desirable conservatism, making a small difference compared to variability within measured storm peak events. This paper also introduces the first publication of a

previously proprietary extended ISO relationship for ambient turbulence intensity:

$$I_u(z) = \left[a_1 \frac{U(z)}{U_{ref}} + a_2 + a_3 \left(\frac{U(z)}{U_{ref}}\right)^{-1}\right] \left(\frac{z}{z_r}\right)^{-0.22} \tag{4}$$

where this time $U(z)$ is taken as the 10 minute mean wind speed, with $U_{ref} = 10$ m s$^{-1}$ and at $z_r = 10$ m. The coefficients $a_1$,

$a_2$ and $a_3$ can be varied to represent site specific data, with default values $a_1 = 0.035$, $a_2 = 0.0089$ and $a_3 = 0.0402$ adopted in the propriety formulation outlined further in the Acknowledgements of this paper. Alternative values of these coefficients will be derived for different atmospheric stability conditions in Sect. 5 of this paper.



Gust factors are the third and final type of vertical profile relationship considered here. They are traditionally used in a
variety of applications, especially estimation of extreme winds for engineering design. They provide an estimate of the most
likely maximum 3 second gust associated with an extreme value of 10 minute or 1 hour mean wind speed. They do not
permit a simulation of all short period variability within the longer averaging period, for which more sophisticated methods
exist. There are opportunities to improve the way extreme gusts are represented in design, beyond the scope of this paper.
The following fixed gust factor is given by IEC 61400-1:2019 and consequently IEC 61400-3-1:2019 for estimating the
maximum 3 second gust associated with a 50 year extreme value of 10 minute mean wind speed:

$$G_{3SEC:10MIN} = 1.4 \tag{5}$$

This is assumed representative of wind turbine hub height but does not vary through the vertical. In other wind engineering
standards such as ISO 19901-1:2015, gust factors $G_{\tau:T}(z)$ vary with height $z$ and are directly related to the turbulence
intensity $I_u(z)$, as formulated below:

$$G_{\tau:T}(z) = 1 - f \cdot I_u(z) \cdot \ln(\tau/T) \tag{6}$$

where $\tau$ is the gust duration and $T$ is the longer averaging period, both in seconds. The widely used Frøya relationship in
ISO 19901-1:2015 includes a coefficient of $f = 0.41$ in Eq. (6) for extratropical storms. The same formulation but with a
coefficient of $f = 0.50$ is cited by Holmes et al. (2007), derived by Ishizaki (1983) for cyclones in Japan. Appropriate values
for this coefficient $f$ will be explored in Sect. 6 of this paper. A range of other gust factor relationships were examined,
including the Wieringa and Mitsuta-Tsukamoto formulations outlined by Emeis (2018). None of these performed as well as
the generalised form of Eq. (6) with either of the above values for coefficient $f$.

Gust factor relationships are often also used to derive ratios for converting estimates of extreme wind speed between
averaging periods much longer than 3 seconds. Following the methods of ISO 19901-1:2015, this would involve using Eq.
(6) with $\tau = 600$ s instead of $\tau = 3$ s and $T = 3600$ s or longer. A range of alternative ratios from IEC 61400-3-1:2019 are
given in Table 1, which do not vary with height. This paper will quantify the performance of these ISO and IEC derived
ratios in Sect. 7, making a clear distinction from the factors used to estimate gusts of a few seconds duration.

**Table 1: IEC ratios for conversion between extreme wind speeds at different averaging periods.**

| Averaging Period (Hours) | Conversion factor relative to extreme 10 minute mean wind speed |
|:---:|:---:|
| 1 | 0.95 |
| 3 | 0.90 |



The ISO and IEC relationships outlined above are generally assumed to be representative of extratropical storm regions, not lower latitude regions impacted by tropical revolving storms. Industry standard relationships applicable to tropical revolving storm conditions can be obtained from the Engineering Science Data Unit (ESDU) models described by Vickery (2014). Similarly, different relationships are applicable to squall winds, as outlined by Santala et al. (2014).

## 2 Offshore Wind Data

The met mast datasets included in this study are summarised in Table 2, broadly in order of descending latitude. At the time of gathering in 2018, the selection included most of the publicly available met mast data in the region, as all long duration FINO met masts were included. Most data are from the public domain, except four confidential datasets made available for these analyses. These include proprietary datasets from Dogger Bank and the original Frøya data, of which the latter were used to derive the ISO relationships of the same name.

**Table 2: Metadata for selected met mast datasets including height of top anemometer above land or mean sea level.**

| Met Mast | Latitude (°N) | Longitude (°E) | Height (m) | Start | End |
|---|---|---|---|---|---|
| Frøya Sletringen | 63.6660 | 8.2590 | 46 | 12-Nov-1988 | 16-May-1989 |
| Frøya Skipheia | 63.6680 | 8.3270 | 100 | 01-Dec-1988 | 16-May-1989 |
| Dogger Bank East | 55.0994 | 2.7025 | 110 | 29-Mar-2013 | 11-Sep-2017 |
| Dogger Bank West | 54.8670 | 1.8200 | 110 | 25-Jul-2013 | 19-Sep-2017 |
| FINO3 | 55.1950 | 7.1583 | 107 | 01-Sep-2009 | 31-Aug-2018 |
| FINO1 | 54.0148 | 6.5876 | 101 | 01-Jan-2004 | 01-Sep-2018 |
| FINO2 | 55.0069 | 13.1542 | 103 | 31-Jul-2007 | 30-Nov-2017 |
| IJmuiden | 52.8482 | 3.4357 | 91 | 01-Nov-2011 | 07-Jan-2016 |
| Egmond aan Zee | 52.6064 | 4.3896 | 116 | 01-Jul-2005 | 31-Dec-2010 |
| London Array | 51.5850 | 1.3940 | 82 | 01-May-2012 | 13-Oct-2014 |
| Kentish Flats | 51.4463 | 1.0781 | 80 | 04-Nov-2002 | 05-Jan-2005 |

The integrity of all measured data was carefully examined with corresponding documentation to remove invalid records. This was a considerable effort, details of which are beyond the scope of this concise paper. Traditional cup anemometers were generally found to be more reliable than sonic sensors. Established methods were used to objectively identify data affected by the mast or other nearby structures, combining anemometers from different orientations at each height. The validity of measured wind direction was therefore a primary consideration, before assessing the validity of wind speeds. Records of adjacent wind farm construction were examined, to identify times and wind directions potentially impacted by wake effects. These were removed before the analysis of clean winds in this paper. Data from the Greater Gabbard met



mast was also gathered, but completely rejected at this stage, because all available data originated from within a constructed

wind farm. Directional screening was also applied to the onshore Frøya met mast data, to only retain winds from the maritime regime. Further intensive data cleaning was conducted iteratively in multiple stages for all met masts, assessing impacts on resulting analyses in extensive sensitivity tests. Proprietary algorithms were developed to identify anomalous profiles, considering the Root Mean Square Error (RMSE) values calculated within the wind shear classification framework described in Sect. 4 of this paper.

Analysis initially focussed on over 1000 storm peak events in the met mast data. These were identified using a peaks over threshold approach, also known as the method of independent storms. These peaks had 10 minute mean winds exceeding a threshold of 20 m s$^{-1}$, separated in time by at least 2 days. Peaks were initially identified at each measurement height individually, then consolidated into a set of vertically coherent peak profile events for each dataset, illustrated in Fig. 1. The height of each peak is not given here, because it varied within each dataset. Several datasets provide good coverage of the

severe winter of 2013-2014, described by Kendon (2015).

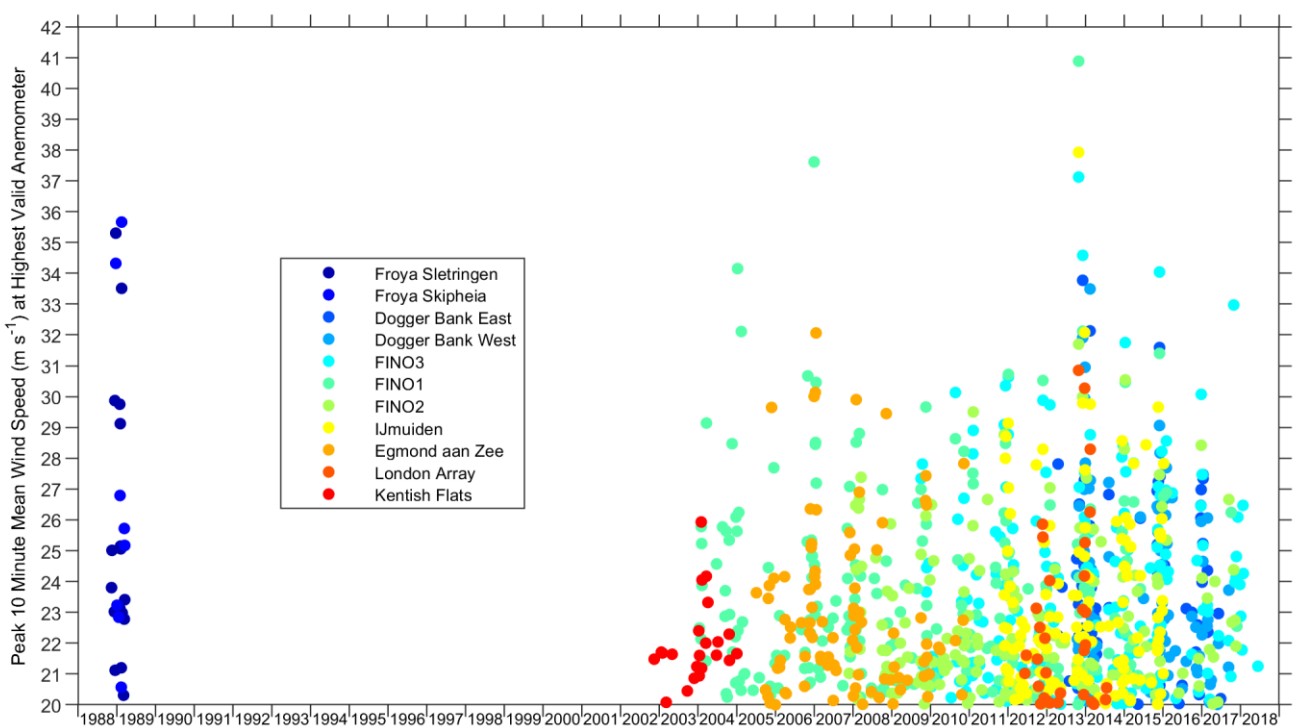

Figure 1: Strongest 10 minute mean winds measured during storm peak events identified at all met masts.



## 3 Wind Profiles and Shear


Measured profiles of 10 minute mean wind speed during the identified storm events of Fig. 1 were compared to the IEC power law of Eq. (1) and ISO Frøya logarithmic profile of Eq. (2). No conversion was made to the 1 hour mean wind speeds in the original Frøya formulation, so this analysis quantifies performance when $U(z)$ and $U(z_r)$ are taken as 10 minute mean wind speeds. Both types of profile were fit to the measured profiles by minimising the RMSE, with the resulting RMSE

values used to quantify performance of the IEC and ISO relationships. The average RMSE, over all peaks with wind speeds over 25 m s$^{-1}$, are shown for each met mast in Fig. 2. Only wind speeds above 25 m s$^{-1}$ are considered, to avoid a higher proportion of profiles with larger errors due to variable atmospheric stability, because ISO Frøya only represents near neutral conditions. No peaks remain above this threshold at Dogger Bank East, where the strongest winds were removed during data cleaning due to anomalies in wind direction. The Frøya advantage was calculated as the difference in average RMSE

and is very close to zero when further averaged over all datasets. This shows very similar performance for both the IEC and ISO profile relationships, suggesting IEC has a very small advantage, much smaller than the variability between the met masts. Some individual wind profiles during key storm events will be examined further in Sect. 4.

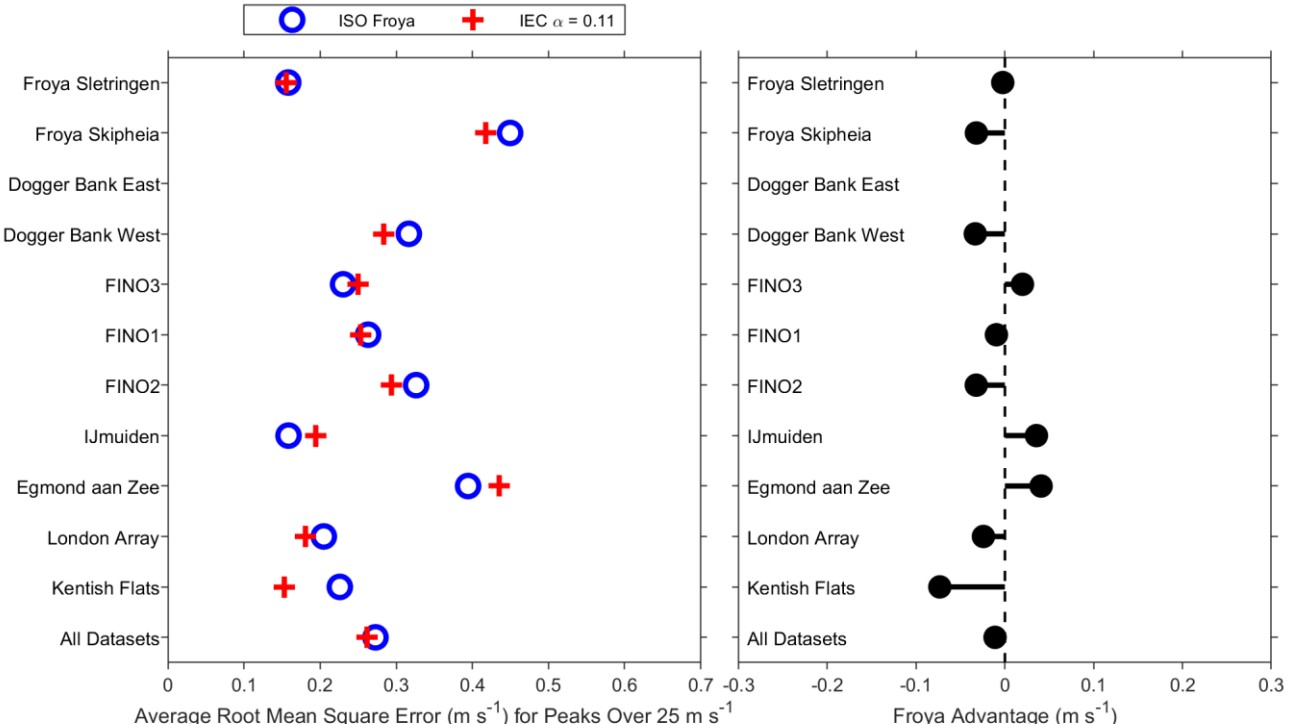

**Figure 2: Quantification of ISO Frøya and IEC profile relationship performance during storm peak events.**





## 4 Relative Shear Classification

The storm peak focussed wind shear analysis of Sect. 3 is now extended to the full range of measured wind speeds. First, the power law exponent $\alpha$ was determined for every valid measured wind speed profile, by minimising the RMSE. The ISO

Frøya profile was similarly fit to every measured profile and the equivalent power law exponent $\alpha_{ISO}$ determined from a further fit to this ISO Frøya profile. A new parameter, the relative shear $r$, was calculated for every measured wind profile by taking the ratio:

$$r = \alpha/\alpha_{ISO} \qquad (7)$$


As indicated by ISO 19901-1:2015, the Frøya profile represents the wind shear associated with near neutral atmospheric stability, which usually dominates in extratropical storm conditions. This paper makes the further assumption that the ISO Frøya profile can be taken to represent wind shear associated with near neutral atmospheric stability across the full range of measured wind speeds, including normal conditions. The relative shear parameter of Eq. (7) can then be used to quantify

how each individual measured profile is impacted by stable or unstable atmospheric conditions.

The above hypothesis was verified using all available measurements of atmospheric stability. A wide range of standard atmospheric stability parameters were estimated from available data and considered at each met mast. An initial shortlist of air-sea temperature difference, vertical temperature gradient and the Monin–Obukhov length were examined. Of these, the clearest trends could be determined using the air-sea temperature difference. This simple measure of atmospheric stability

was also found to be the most effective in the same region by Furevik and Haakenstad (2012) then again by Kettle (2014).

The calculation of relative shear in Eq. (7) and relationship with measured atmospheric stability is shown in Fig. 3 for the FINO1 met mast, with all other relevant met mast analyses shown in Appendix A. FINO1 is shown here because it was the longest duration of all datasets considered in this paper. The scatter plots show all measured wind profiles (within the axes limits) with shading used to indicate data density. Trends can be clearly seen in the measured mean values in each bin of the

horizontal axes. As found in Sect. 3, the performance of IEC $\alpha = 0.11$ and ISO Frøya $\alpha_{ISO}$ are practically indistinguishable at the strongest measured wind speeds, but ISO Frøya $\alpha_{ISO}$ gives a reasonable approximation of mean conditions across the full range of normal wind speeds.

There is a clear correlation between relative shear and atmospheric stability measured by air-sea temperature difference. This follows the expectations of shear computed via Monin–Obukhov theory, which gives linear wind speed profiles in cases

of very strong stability. These cannot be accurately represented by power law profiles or consequently by the relative shear parameter introduced here. Simple linear slope fits were therefore also made to all measured wind profiles, with positive slopes representing strong stability. Negative linear slopes were also included, to capture the inverted profiles documented in the region by Kettle (2014).



**Figure 3: Relative shear and atmospheric stability analysis for FINO1 met mast.**


**Figure 4: Relative shear and linear slope analyses for a selection of measured wind profiles.**



The RMSE values from the power law and linear slope fits were compared and linear fits were preferred when RMSE was

lower by at least 0.066 m s$^{-1}$. This somewhat arbitrary threshold was determined through iteration, to retain the traditional power law fit when small errors could otherwise lead to a linear fit being selected. The profile classification method is illustrated in Fig. 4. The ISO Frøya profile and equivalent $\alpha_{ISO}$ are shown in grey, with the power law and linear fits in blue and red respectively. The RMSE is given in the colour corresponding to the preferred fit. The upper subplots show profiles at the peak of the strongest storm event in the collated data, as it impacted four met masts. These storm peak profiles were

classified by the relative shear values in blue. The lower two subplots of Fig. 4 show profiles that were better classified with positive or negative linear slopes, with values shown in red. The strongly stable profile from IJmuiden was among the independent storm peak events identified and examined in detail at this met mast.

Thresholds of relative shear were selected to classify profiles as unstable, neutral or stable. These were embedded within a wider wind shear classification framework including the negative (inverted) and positive (strong stability) linear slopes,

leading to the wind profile clustering in Fig. 5. The horizontal axes show fixed thresholds used to classify neutral conditions and the maximum relative shear before positive linear slopes prevailed. The terminology relative shear classification is used in this paper, to make it clear that a traditional classification according to atmospheric stability has not been performed. However, the new framework provides a pragmatic proxy for such a stability classification, used in Sect. 5 to obtain different ambient turbulence intensity relationship for distinct stability conditions.

## 5 Turbulence Intensity


This paper now quantifies the performance of the turbulence intensity relationships in Sect. 1, with some other relationships from IEC standards also briefly considered. The focus is ambient turbulence intensity, representing clean wind conditions in the absence of wake effects from near or far wind turbines. Analysis is based on measured turbulence intensity $I_u(z)$ derived from the standard deviation and corresponding 10 minute mean wind speeds from each anemometer on each met mast. By

far most of the data retained for this analysis were derived from traditional cup anemometers. The 90 percentile (P90) and mean values of turbulence intensity were derived for each bin in 10 minute mean wind speed. Results are shown from the measurement height closest to 80 m in Fig. 6 and Fig. 7. This height selection gave the highest valid data at FINO1 during the strongest measured storm and generally good data coverage at all met masts.

The 90 percentile values in Fig. 6 are compared with the IEC 61400-1:2019 Normal Turbulence Model (NTM), which is

prescribed for onshore conditions, but still referenced in IEC 61400-3-1:2019 for offshore turbines. Fig. 6 also shows offshore relationships from IEC 61400-3-1:2019 and the earlier IEC 61400-3:2009. These offshore IEC relationships are derived from the same equation, in which only one parameter (d) changed between 2009 and 2019. The roughness length was taken as the value needed to match a fit to the ISO Frøya logarithmic profile of Eq. (2), not the implicit Charnock relationship in the offshore IEC standards. In all the above IEC relationships, the reference value of turbulence intensity was

taken to be 0.12, for the least turbulent wind turbine class C.



**Figure 5: Percentage occurrence histograms for relative shear and linear slope wind profile classes.**





**Figure 6: Comparison of measured 90 percentile turbulence intensity with standard relationships.**







**Figure 7: Comparison of measured mean turbulence intensity with standard relationships.**



Results from the relationship of Wang et al. (2014), referred to by IEC 61400-3-1:2019, are also shown in Fig. 6. The default parameters in the publication were used, but optimisation of these to represent site specific measured data is recommended. This is not performed in this paper, which now moves on from 90 percentile turbulence intensity to consider

how the ISO and extended ISO relationships from Sect. 1 represent mean turbulence intensity and values from the peak of each storm. These are compared in Fig. 7, which also shows values from the onshore IEC 61400-1:2019 NTM, because this is the only IEC relationships which permits mean turbulence intensity to be calculated directly.

Fig. 7 shows the extended ISO relationship from Eq. (4) with the default coefficients given in Sect. 1, which clearly provides the best representation of measured turbulence intensity of all the relationships considered so far. The extended ISO

relationship is much better than the ISO Frøya relationship of Eq. (3) at low wind speeds. However, it is unclear which of these provides superior representation at high wind speeds, especially when the individual peak values are considered. The most severe storm event in the collated data, shown in Fig. 4, led to the strongest overall wind speeds at several met masts. This is visible as an isolated grey dot between the ISO Frøya and extended ISO lines for FINO1 in Fig. 7. It is useful to consider the scatter in the storm peak values of turbulence intensity and how this relates to an assessment of uncertainty.

The ISO Frøya turbulence intensity values in Fig. 7 were calculated with Eq. (3) taking $U(z_r)$ as the 1 hour mean wind speed at $z_r = 10$ m, but plotted versus the corresponding 10 minute mean wind speed at measurement height. The values would be slightly higher if $U(z_r)$ was taken as the 10 minute mean wind speed at $z_r = 10$ m, such that the difference would hardly be visible in Fig. 7 and certainly much smaller than the scatter in individual peak values.

The relative shear and linear slope classification framework described in Sect. 4 is now used to cluster the measured

turbulence intensity data, broadly representing different types of atmospheric stability. A further distinction was made between maritime and terrestrial regimes, according to the wind direction associated with each measured profile. The maritime regime was defined as direction sectors with a distance from the coast of at least 50 km. This followed Pollak (2014), who found turbulence intensity decreased with increasing distance from the coast until about this value.

The coefficients of the extended ISO relationship were modified to provide an improved representation of measured

turbulence intensity in each relative shear class, shown for the maritime regime in Fig. 8 to Fig. 12. Key features of the measured data could be effectively characterised by only two sets of coefficients, given in Table 3. These are denoted the ENOW coefficients, after the Joint Industry Project on which they were derived. A notable feature of this analysis is how the same relationships can represent turbulence intensity at most met masts, supporting the hypothesis that there may be less uncertainty and spatial variability in this parameter than sometimes assumed. The corresponding analyses for the terrestrial

regime are shown in Appendix B. These use the same modified coefficients in each relative shear class as the maritime regime, because to a first approximation they provided a reasonable representation of measured conditions.







**Figure 8: Modification of extended ISO relationship for maritime inverted profiles.**



**Figure 9: Modification of extended ISO relationship for maritime unstable profiles.**








**Figure 10: Modification of extended ISO relationship for maritime neutral profiles.**





Figure 11: Modification of extended ISO relationship for maritime stable profiles.





**Figure 12: Modification of extended ISO relationship for maritime linear profiles.**





**Table 3: ENOW coefficients of the extended ISO relationship for maritime regime ambient turbulence intensity.**

| Relative Shear Class | $a_1$ | $a_2$ | $a_3$ |
|---|---|---|---|
| Inverted | 0.025 | 0.0450 | 0.030 |
| Unstable | 0.025 | 0.0450 | 0.030 |
| Neutral | 0.025 | 0.0450 | 0.030 |
| Stable | 0.035 | 0.0089 | 0.027 |
| Linear | 0.035 | 0.0089 | 0.027 |


## 6 Gust Factors

This paper now reverts to the initial focus on the storm peak events considered in Sect. 3, considering the performance of the gust factor relationships of Sect. 1. Measured gust factors were derived directly from the 10 minute data or by taking the 1 hour mean wind speed centred on each 10 minute peak. Eq. (6) was used to calculate gust factors at all measurement

heights, for a range of values of coefficient $f$. Importantly, this calculation used the measured turbulence intensity described in Sect. 5, not estimates of $I_u(z)$ derived from any of the relationships in Sect. 1. The percent error in gust factor was calculated at each measurement height, then averaged through the vertical. Further averages were then taken over all peak events at each met mast, to produce the summary in Fig. 13. Detailed assessment of over 1000 storm peak profiles confirmed this concise summary represented trends at all measurement heights.

The horizontal axis in Fig. 13 does not indicate the gust duration, which varied between met masts and was usually 1 or 3 seconds. Met masts with 3 second gusts are indicated by a corresponding IEC value from Eq. (5) on the left on Fig. 13. These have a consistent large positive percent error, indicating the IEC gust factor provides a conservative overestimate of extreme 3 second gusts. The Frøya relationship is superior to Ishizaki for estimating gusts from 10 minute mean wind speeds. An intermediate value of $f = 0.46$ on the right of Fig. 13 provides the best overall performance for estimating gusts

from 1 hour mean wind speeds.

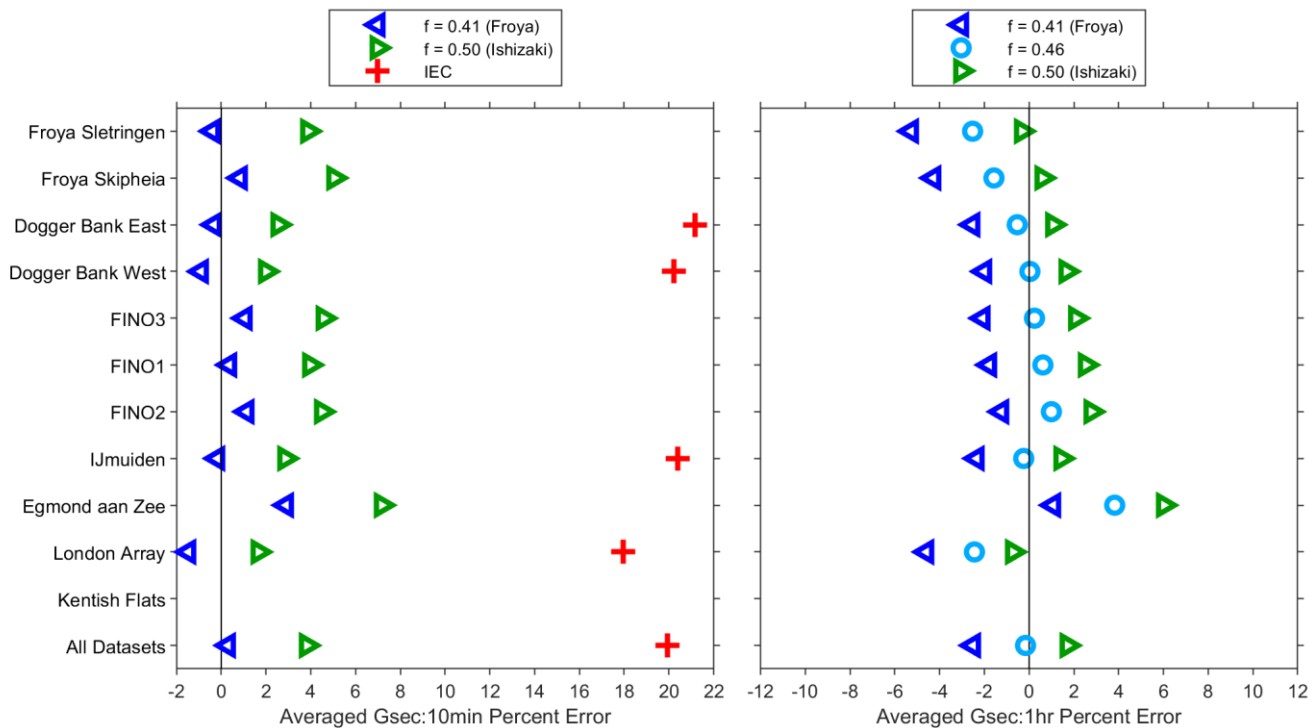

**Figure 13: Averaged percent error in estimates of gust factor during storm peaks from all met masts.**

## 7 Averaging Period Ratios

The gust factors considered in Sect. 6 are distinguished from the ratios used to convert extreme winds speed estimates from one averaging period to another, as explained in Sect. 1. The performance of ratios derived from ISO Eq. (6) and IEC Table 1 are now assessed, using ratios between the measured 10 minute and 1 hour mean wind speeds described in Sect. 6. New measured 3 hour mean wind speeds centred on each 10 minute peak are now also included in the analysis. Percent errors were again averaged through the vertical then over all peaks at each met mast to produce the summary in Fig. 14.

The IEC ratios from Table 1 exhibit similar performance to those derived from ISO Eq. (6). As in Sect. 6, ratios derived from additional values of the coefficient $f$ are shown when they provide the best overall performance. This suggests the best estimates of peak 10 minute mean wind speeds are obtained from 1 hour mean values using $f = 0.45$, or from 3 hour mean values using $f = 0.53$.

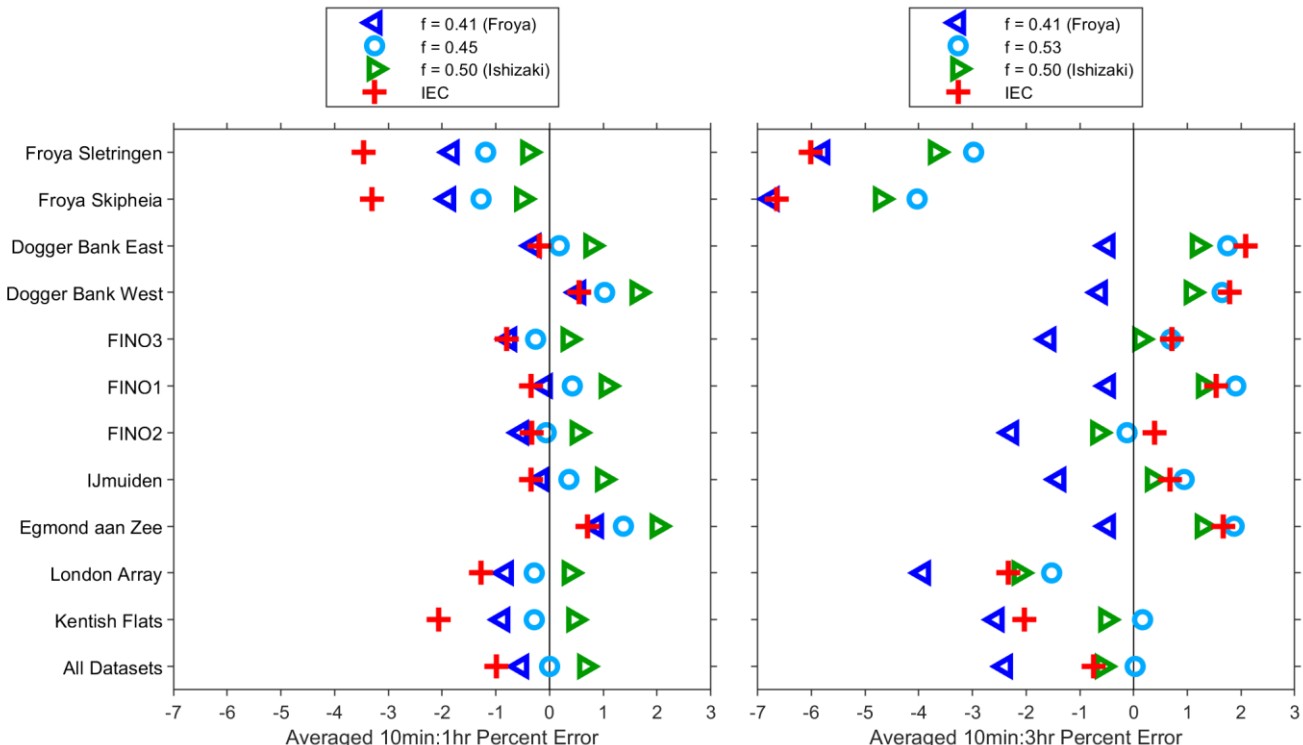


**Figure 14:. Averaged percent error in ratios to estimate 10 minute mean peak values from longer averaging periods.**

## 8 Extreme Wind Workflow

Relevant findings from previous parts of this paper are now brought together to demonstrate application in a simplified workflow for estimating extreme wind speeds for offshore wind energy applications. This starts with the assumption that a
reliable estimate of 50 year extreme 10 minute mean wind speed is available at 100 m above the sea surface. The origin of the initial estimate is beyond the scope of this paper. Subsequent steps in the workflow are illustrated in Fig. 15, with the sequence indicted from top to bottom in the legend.

First, the Frøya relationship of Eq. (2) is used to derive a coherent vertical profile of extreme 10 minute mean wind speed, noting that this not necessary assumed valid above the height of traditional met masts. Fig. 15 indicates that the shear is only
a little greater than IEC $\alpha = 0.11$ in Eq. (1), so that could be a pragmatic alternative. Next, the extreme 10 minute mean wind speed at 10 m is used to estimate a profile of turbulence intensity using ISO relationship in Eq. (3). The value of $I_u(z)$ at 100 m above sea level is then used to estimate extreme 1 hour mean wind speed at 100 m via the ISO gust factor relationship of Eq. (6), in this case using the default $f = 0.41$. Fig. 15 shows the corresponding ratio is close to the IEC value in Table 1, which again offers a pragmatic alternative. Finally, a coherent vertical profile of extreme 1 hour mean
wind speed is obtained using the ISO Frøya relationship of Eq. (2). Again, the shear is a little greater than IEC $\alpha = 0.11$.





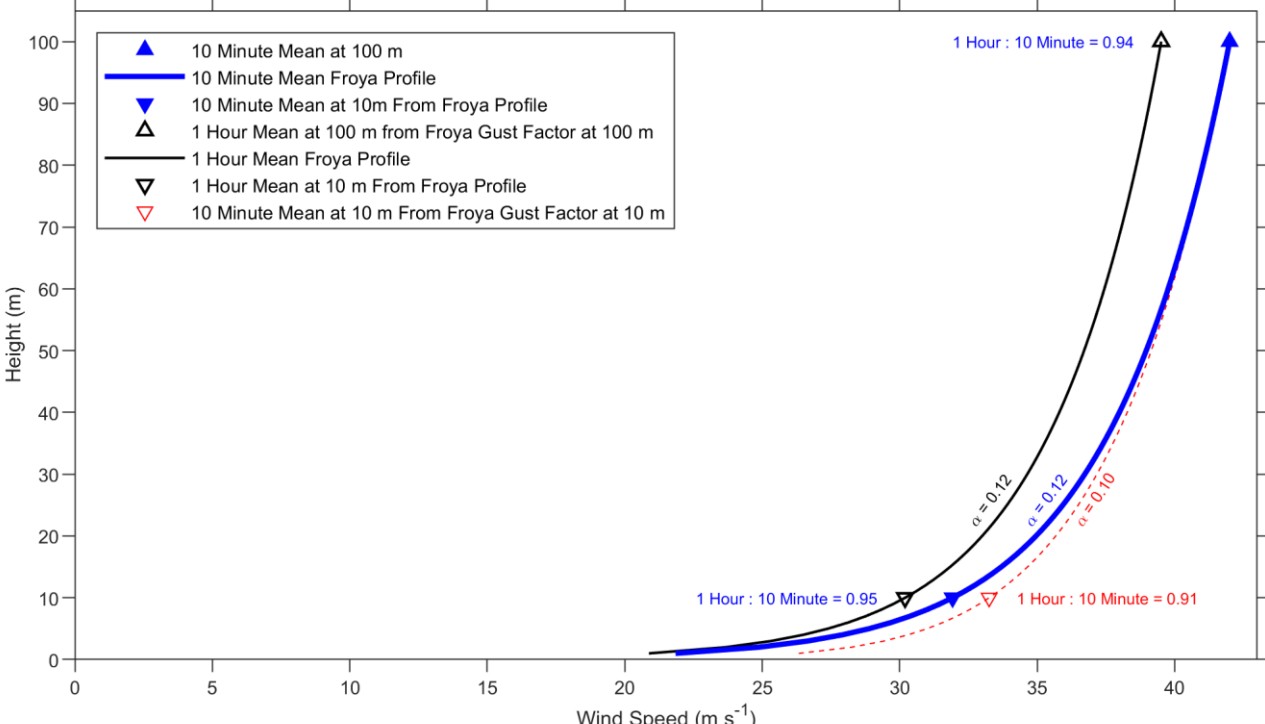

Figure 15: **Extreme wind workflow with alternative estimates of 10 minute mean winds at 10 m.**

The workflow outlined above differs from methods commonly adopted by some metocean practitioners, which usually need to be independently reviewed and often certified. In metocean practice, the original ISO relationships are often used more literally, with an emphasis of obtaining an estimate of 1 hour mean wind speed at 10 m earlier in the workflow, as a key input to Eq. (3). However, the workflow can be streamlined by using some of the pragmatic simplifications explored in this paper, which in some cases might involve using the much simpler IEC alternatives. A robust assessment of uncertainty has

not yet been undertaken by the author, but the differences between these shortlisted methods are expected to be small compared to other uncertainties involved in the estimation of extreme wind speed profiles.

An alternative estimate of 10 minute mean wind speed at 10 m is shown in red in Fig. 15, representing part of the widely used metocean tradition following ISO. This is derived from the 1 hour mean wind speed at 10 m via the turbulence intensity at 10 m, which is higher than the turbulence intensity at 100 m according to Eq. (3). The gust factor from Eq. (6) is

therefore higher at 10 m than 100 m, indicted by the different ratio shown in red in Fig. 15. This higher red estimate of 10 minute mean wind speed at 10 m is interpreted as the value *associated with* the 1 hour mean wind speed at 10 m. In contrast, the lower blue estimate of 10 minute mean wind speed at 10 m is interpreted as the value *associated with* the 10 minute mean wind speed at 100 m. This latter value is more consistent with IEC and represents a coherent vertical profile, arguably more appropriate for wind turbine engineering.





## 9 Conclusions


The IEC $\alpha$ = 0.11 power law and ISO Frøya profiles proved practically indistinguishable for quantification of shear during measured storm peak events. Either can be used for related applications, although the ISO Frøya relationship has the advantage of wind speed dependence. It is important to highlight how the shear analysis in this paper only considers the atmospheric boundary layer up to heights covered by traditional met masts. Use of LiDAR data to assess how high the IEC

and ISO wind profile relationships can be practically extended during storm peak events is beyond the scope of this paper. The reliability of LiDAR data under strong wind conditions must be considered carefully for such applications.

The relative shear analysis in this paper integrates the IEC and ISO preferred methods of wind profile characterisation. The established ISO Frøya logarithmic profile is used to quantify neutral conditions, from which IEC favoured power law profiles are rescaled in terms of relative shear. Clear correlations permit this to be used as a practical approximation of

atmospheric stability. Relative shear is embedded within a wider classification framework, incorporating positive and negative linear slopes, to represent strongly stable and inverted profiles respectively. This offers a simple alternative to Monin–Obukhov theory for representing strong stability, while capturing the inverted profiles that Monin–Obukhov theory cannot. This new classification framework requires no measurement of atmospheric stability, so can be applied directly to any offshore LiDAR dataset.

This paper proposes just two fundamentally different offshore turbulence intensity relationships, distinguishing unstable or neutral conditions from stable conditions. In the absence of wake effects, offshore turbulence intensity is clearly lower than prescribed by even the weakest IEC turbine class. To first approximation, there is little difference between maritime and terrestrial regimes. This is a pragmatic simplified analysis and opportunities exist for further refinement using the wealth of available data. Such developments are underway, aiming to investigate fetch, directional, seasonal and vertical trends.

Quantification of 90 percentile rather than mean turbulence intensity is required for some wind energy applications. The coefficients of Wang et al. (2014) could be modified as were those of extended ISO. The quantification of turbulence intensity from fixed or floating LiDAR data remains the subject of ongoing research.

Most measured storm peak profiles fall into the neutral relative shear classification, but not all. This reflects the somewhat arbitrary thresholds used to distinguish these classes. A wider definition of near neutral conditions would include more of the

measured peak events. Mean turbulence intensity under neutral conditions in Fig. 10 is consistently a little higher than given by the traditional ISO Frøya relationship, reflected by the extended ISO modifications. This further supports the pragmatic use of $U(z_r)$ as the 10 minute mean wind at $z_r$ = 10 m in Eq. (3). This would increase turbulence intensity above the original 1 hour mean wind derivative by a factor of about 2%, which would be difficult to see in Fig. 10 and much smaller than the scatter in measured peak values. Mean turbulence intensity is similarly higher than ISO Frøya in unstable

conditions, but clearly lower in stable conditions, in line with physical expectations.

The ISO Frøya gust factor relationship is recommended for estimating the most probable 3 second gusts associated with extremes in 10 minute mean winds. A modified coefficient $f$ = 0.46 in Eq. (6) is superior when these gusts are estimated



from 1 hour mean winds. Further modifications of this coefficient in Fig. 14 are effective when Eq. (6) is used to estimate peak values of 10 minute mean wind from peaks representing averaging periods of 1 or 3 hours. However, the IEC ratios of

Table 1 are a simple pragmatic alternative, with percent errors comparable to the variability between individual met masts. There is greater variability and uncertainty in these peak wind speed conversion factors when the difference between the averaging periods is larger. This can to some extent be attributed to greater nonstationarity within longer averaging periods. A few of the strongest peak events were strongly nonstationary and deviated from the most probable trends quantified here. The quantification and engineering impacts of strongly nonstationary events, such as sharp fronts or squalls, remains a

subject of active research. This may be increasingly relevant to floating offshore wind structures.

This study has focused on quantifying the most probable 3 second gusts associated with extremes in 10 minute or longer averages in wind speed. The IEC extreme gust factor of Eq. (5) is clearly higher than measured in most peaks. However, different extreme gust estimates can be derived by considering the distribution all gusts within each storm, rather than only those occurring at the same time as peaks in wind speed averaged over longer periods. Related research involving this

author will soon be submitted for publication in a companion paper.

Efforts have been made to disseminate useful results from extensive data analyses in a concise coherent paper. Details of wind shear are presented for all relevant measured profiles. Turbulence intensity is presented as statistics over the full range of wind speeds, plus individual values from each storm peak event. Statistical results are even more condensed for gust factors and averaging period conversions, but all values from individual storm peaks were assessed before developing a

representative summary of the key trends. This paper does not attempt to include sufficient detail on all variability to permit a robust assessment of uncertainty.

The storm peak focus of this paper differs from numerous other published studies. Some of the relationships examined literally converge at high wind speed, but related research addresses a wider convergence of technical disciplines within the rapidly growing offshore wind industry. The ISO Frøya relationships were originally derived from the limited data over land

that were included in this study. These relationships are now validated against much longer duration data, covering a considerably wider offshore region more relevant to wind energy. They are used extensively in offshore engineering, but not yet widely adopted in offshore wind energy. This paper is intended to demonstrate some benefits of appropriately incorporating these relationships into industry standard practice. Key benefits include performance in directly representing profiles of 10 minute mean wind and providing a pragmatic method for approximating near neutral wind shear. The extreme

wind workflow outlined in this paper is intended to explore how established ISO and IEC can be practically combined and stimulate further debate within the wind energy community.



## Appendix A

The calculation of relative shear and correlation with atmospheric stability described in Sect. 4 is shown in Fig. A1 to Fig.
A6 for all met masts at which air-sea temperature difference was measured.

## Appendix B

The modified coefficients of the extended ISO relationship in each relative shear class for the terrestrial regime are shown in
Fig. B1 to Fig. B5.

Figure A1: Relative shear and atmospheric stability analysis for Frøya Sletringen met mast.

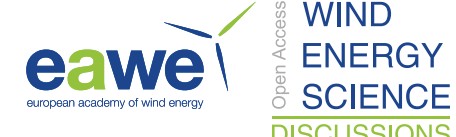

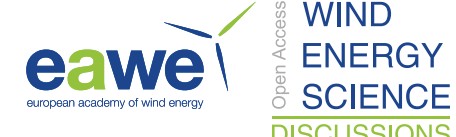

**Figure A2: Relative shear and atmospheric stability analysis for Dogger Bank West met mast.**



**Figure A3: Relative shear and atmospheric stability analysis for FINO3 met mast.**

**Figure A4: Relative shear and atmospheric stability analysis for IJmuiden met mast.**




**Figure A5: Relative shear and atmospheric stability analysis for Egmond aan Zee met mast.**

**Figure A6: Relative shear and atmospheric stability analysis for Kentish Flats met mast.**



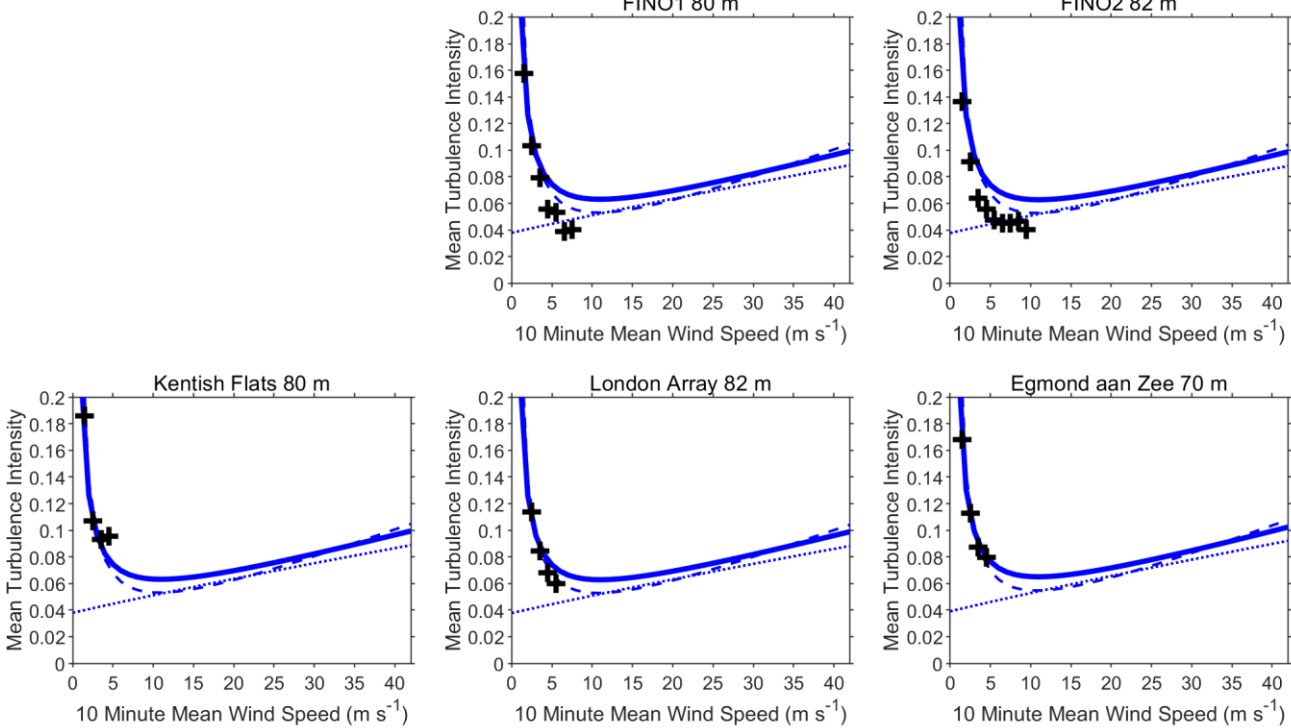

**Figure B1: Modification of extended ISO relationship for terrestrial inverted profiles.**



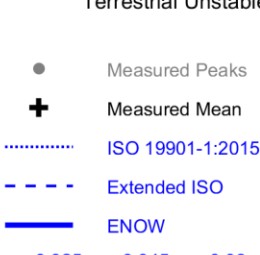

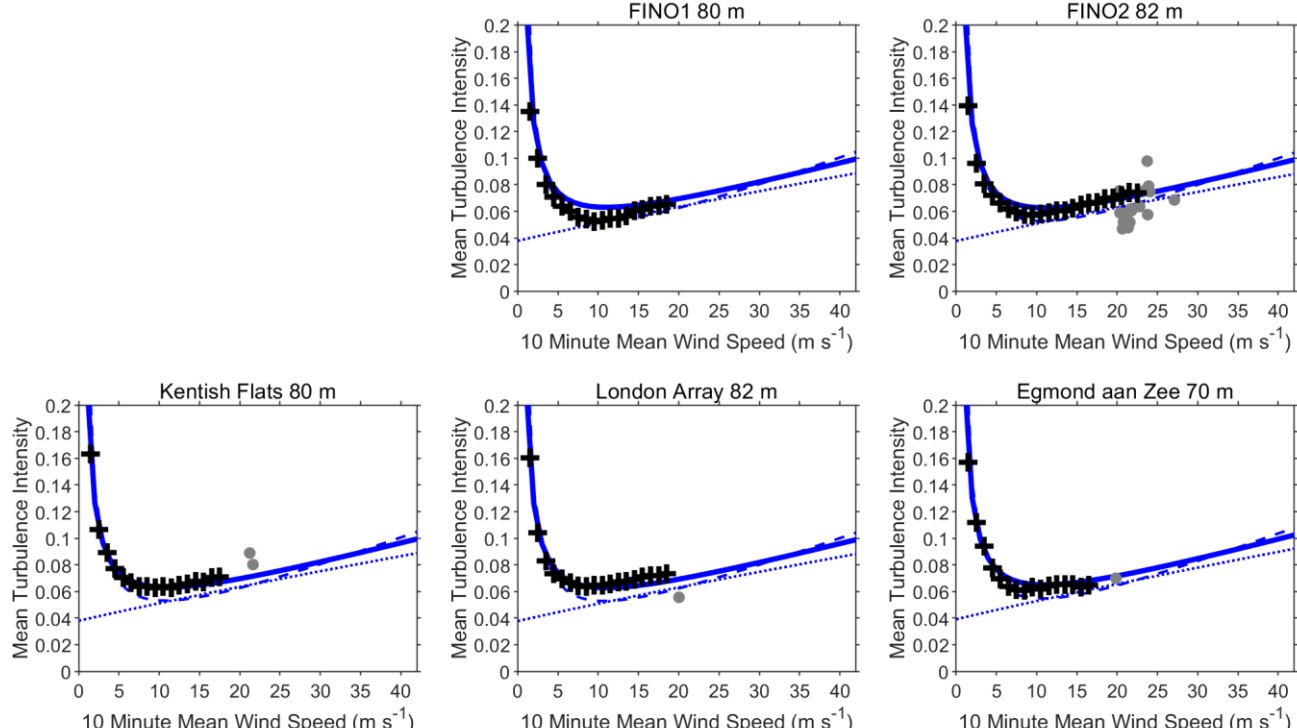

**Figure B2: Modification of extended ISO relationship for terrestrial unstable profiles.**



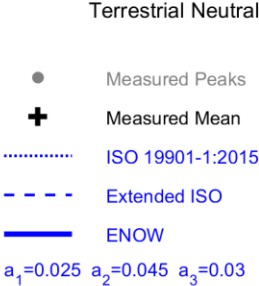

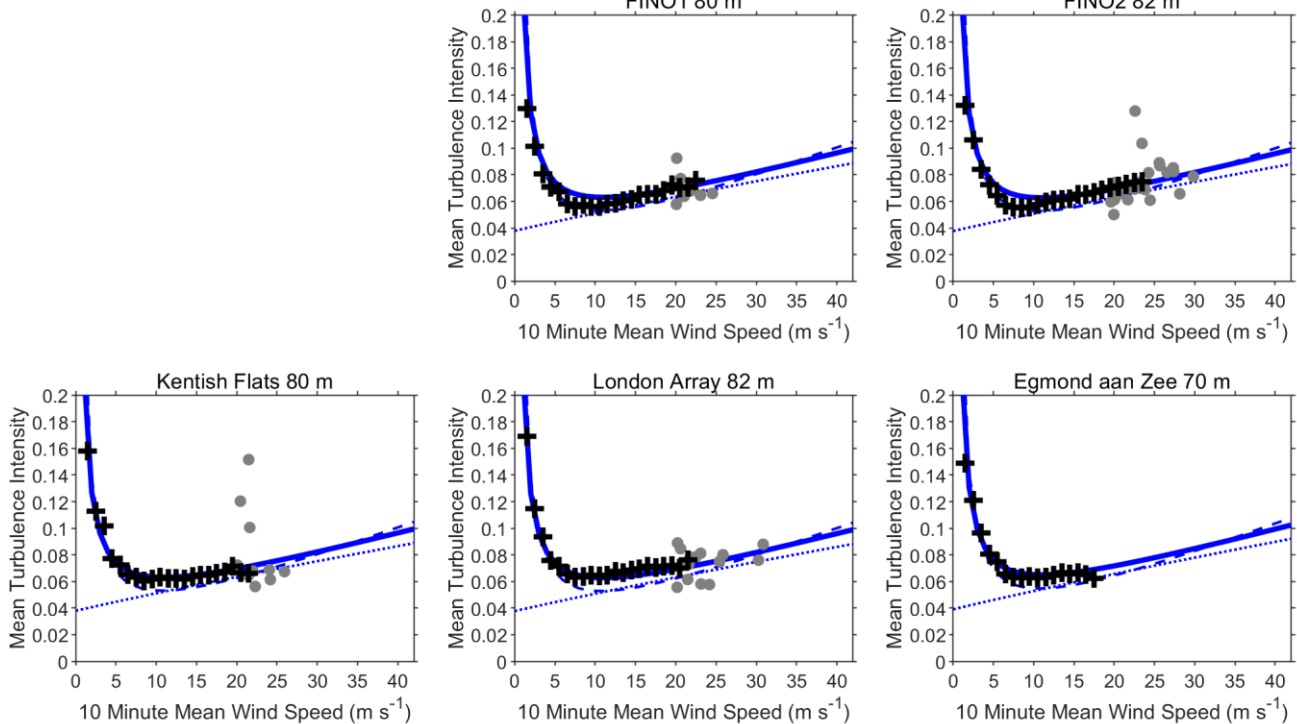

**Figure B3: Modification of extended ISO relationship for terrestrial neutral profiles.**





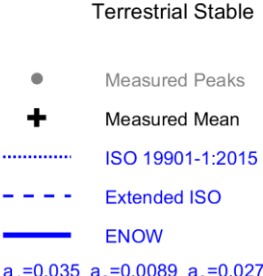

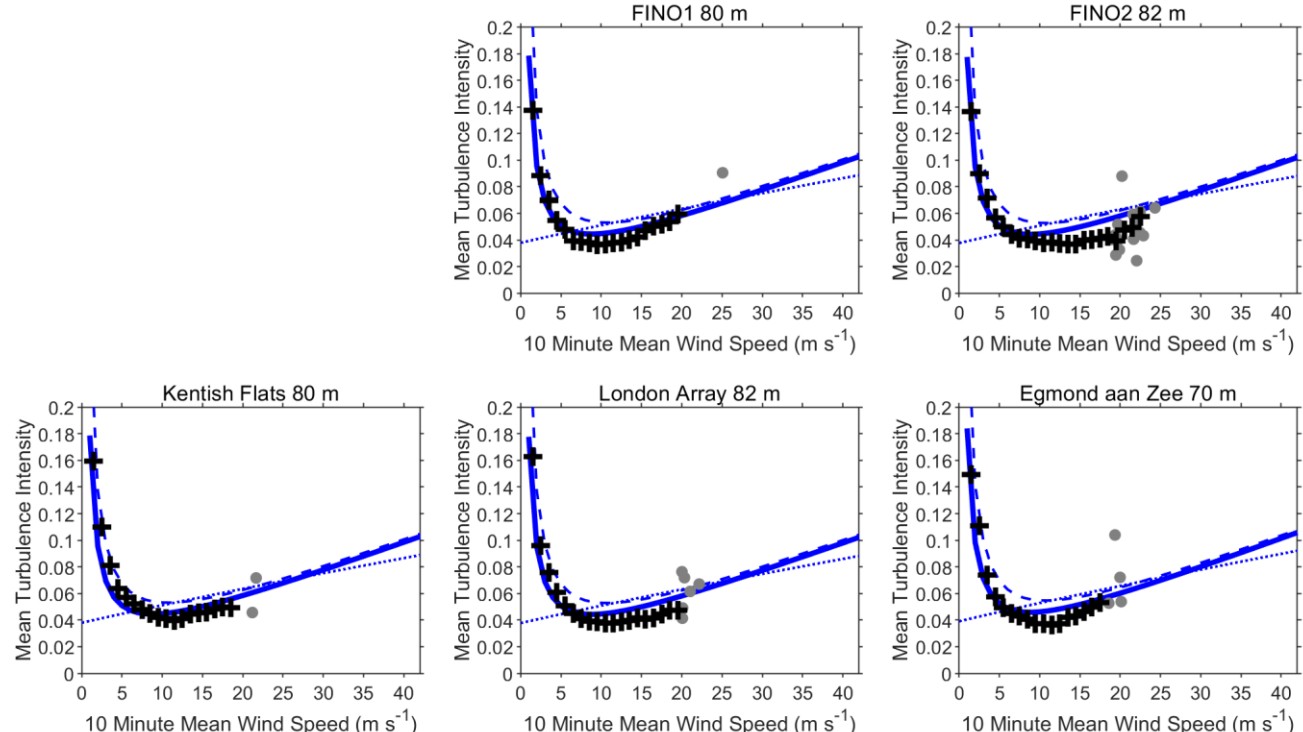

**Figure B4: Modification of extended ISO relationship for terrestrial stable profiles.**




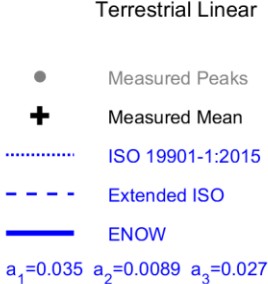

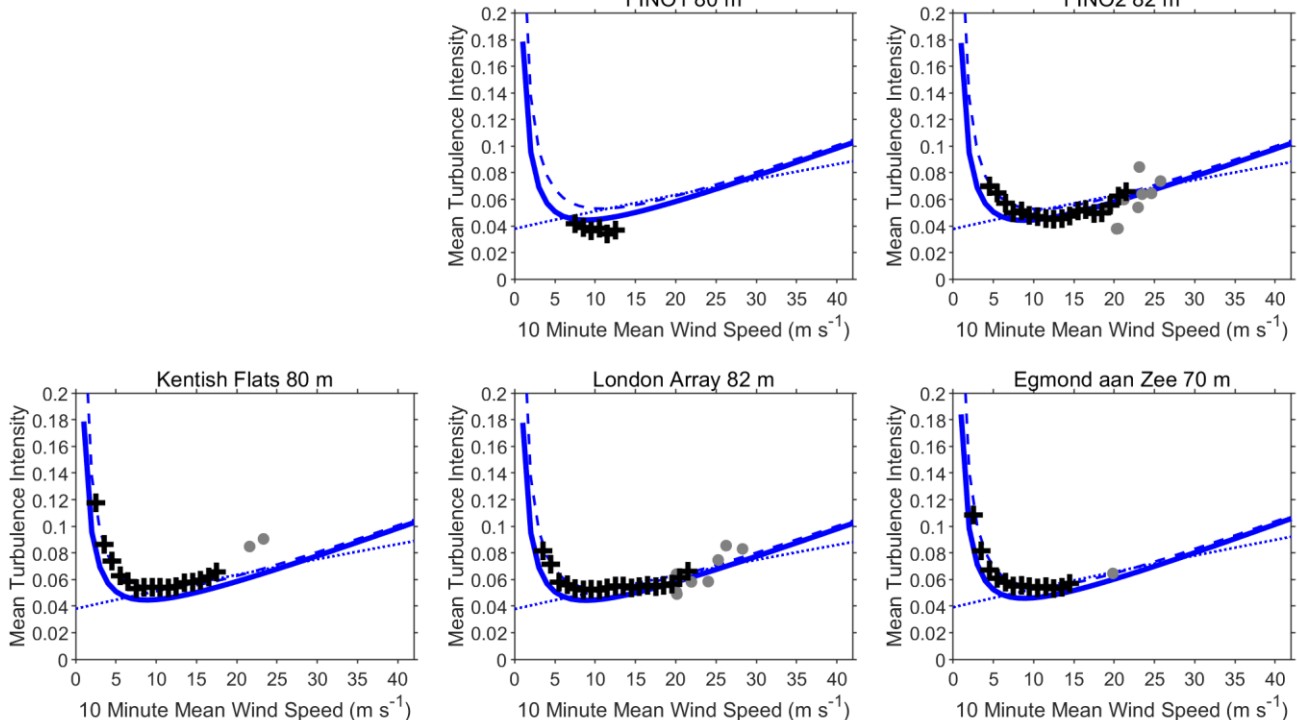

**Figure B5: Modification of extended ISO relationship for terrestrial linear profiles.**



**Author Contribution**

This paper is the work of the author, in some cases building on contributions from others named in the Acknowledgements.

**Competing Interests**

No conflicts of interest exist, but it is desirable to clearly state how this applied research was funded and delivered by commercial organisations. The author works as a metocean consultant in the offshore wind energy industry and this paper is derived from confidential results of the Extreme and Normal Offshore Wind (ENOW) Joint Industry Project (JIP). The author is also an active member of the PEL/88 committee for UK contributions to IEC standards on wind energy generation systems and initiated the Wind Resource Metocean (WRM) discussion group. The objective of this paper is to openly share meaningful results within the wind energy community, to support wider efforts to reduce uncertainty and promote engagement between related technical disciplines.

**Acknowledgements**

In order of joining the project, the following ENOW participants are thanked for financial support and permission to publish: Vattenfall, Shell, Equinor, TotalEnergies, bp and SSE Renewables. Among numerous participant representatives, Einar Nygaard, Zoe Roberts, Paul Verlaan and Alison Brown are acknowledged for helping to create the project. Andrew Watson and Dave Quantrell made notable contributions at an early stage of the project, leading to some of the results presented here. Laure Grignon and Gil Lizcano are also thanked for valuable early contributions. The Extended ISO relationship described in this paper was provided for consideration in the ENOW project by Equinor in a confidential report by Martin Mathiesen. The original Frøya data and both Dogger Bank datasets were provided in confidence by Equinor to support the ENOW project. Data from all three FINO masts were acquired from the German Federal Maritime and Hydrographic Agency (BSH), with support from the BMWi (Bundesministerium fuer Wirtschaft und Energie, Federal Ministry for Economic Affairs and Energy) and the PTJ (Projekttraeger Juelich, project executing organization). The IJmuiden dataset was acquired from www.WindOpZee.net via Energy research Centre of the Netherlands (ECN). The Egmond aan Zee data were downloaded from the NoordzeeWind web pages. The London Array and Kentish Flats data were acquired via the UK Crown Estate Marine Data Exchange (MDE).





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
