# Peer review of "Converging Profile Relationships for Offshore Wind Speed and Turbulence Intensity"

_Wind Energy Science, 2023_

## Referee Comment (RC2)

[referee-annotated manuscript omitted]

---

## Community Comment (CC3)

[revised manuscript text omitted]

---

## Author Response (AR1)

**Revisions to Converging Profile Relationships for Offshore Wind Speed and Turbulence Intensity**
Gus Jeans
May 2024

**Response to First Formal Review**

The suggested reasons for rejection given by the anonymous first formal reviewer have already been robustly rebutted in the public discussion[1]. This is firmly supported by all three subsequent and very relevant participants in the peer review[2][3][4]. I now add the following to my response to each numbered point made by this reviewer:

1. I have checked the definition of science provided by the Science Council[5], with whom I am professionally registered as Chartered Scientist: "*Science is the pursuit and application of knowledge and understanding of the natural and social world following a systematic methodology based on evidence*". I am confident that the work presented in this paper clearly meets this widely accepted definition. Furthermore, it should be recognised that engineering is a branch of applied science and the results in this paper already have recognised practical value to the wind energy industry.

2. I have clarified the need for LiDAR data in the conclusions. I have already extended the shear analysis to higher elevations using LiDAR, but these results are not approved for publication by the ENOW participants and would be best presented in a separate paper if they were available. However, shear is a small part of this research, while LiDAR is not presently accepted to quantify turbulence intensity or gusts.

3. No further response is needed. The published analyses are already systematic, evidence based and directly applicable to many offshore wind energy development projects.

**Response to Second Formal Review**

I very much welcome these comments which have led to a more digestible publication. The paper has been substantially shortened and focussed as suggested in the first comment by the second formal reviewer. These changes closely follow the suggestions provided in the annotated PDF, as detailed on the following page. I have removed all reference to the P90 representative turbulence intensity because this analysis was incomplete at the time of writing and further results remain unavailable for publication. This removes the need to add some of the detail requested in the second comment by this reviewer, supporting efforts to shorten and focus the paper.

**Other Changes**

I have taken this opportunity to make a few other edits that can all be seen in the submitted tracked changes document. Key changes are summarised here.

I corrected the blue ISO line in Figure 3 which was erroneously plotted versus wind speed at 10m not 100m. This cannot be directly compared to the preprint, as the overall figure has been restructured according to review comments.

I added a short paragraph about limits to the drag coefficient at very high wind speed, with a corresponding reference, following some very interesting discussions with a colleague. I also fixed the reference to Holmes (2007).

I updated my related professional activities in the competing interests section, including IEC-ISO liaison. I cited this paper as very relevant when volunteering for this role, so I am keen to complete publication soon!

Finally, I have added a family dedication to the acknowledgements, as these big life and death events became entangled with the difficulties encountered with this publication, making everything require much more time.
* * *
[1] https://doi.org/10.5194/wes-2023-35-AC1

[2] https://doi.org/10.5194/wes-2023-35-CC1 with reply https://doi.org/10.5194/wes-2023-35-AC2

[3] https://doi.org/10.5194/wes-2023-35-CC2 with reply https://doi.org/10.5194/wes-2023-35-AC3

[4] https://doi.org/10.5194/wes-2023-35-CC3 with reply https://doi.org/10.5194/wes-2023-35-AC4

[5] https://sciencecouncil.org/about-science/our-definition-of-science/

**Detailed Response to PDF Comments from Second Formal Review**

Page 5: I added the recent Dogger Bank addition to MDE in the acknowledgements, which I believe already includes sufficient details of all data sources for a concise paper. Regarding offshore exposure, this varies considerably with direction and all masts were offshore except Froya, as now clarified in the table caption. Finally, mASL will not be added as the reference level varies and is already given in the table caption.

Page 8: This section is considerably shortened as suggested with details provided in the tracked changes document. This includes removal of Appendix A. I have not removed all the text suggested by the reviewer, as this would remove one of the references and introduction of Figure 3.

Page 9: Figure 3 restructured as suggested.

Page 10: Wind direction profiles have not been added as this would conflict with efforts to shorten the paper. But more importantly, no new ENOW derivatives can be added to the paper at this stage, because it would require another time consuming formal review and approval by all six major wind energy development companies who support the JIP. I do have their permission to shorten the paper without further review. In figure 4, I have retained the event with winds less than 25 m/s, this is important to illustrate representation of an inverted profile via negative linear slope.

Page 11: As page 8, this section is shortened as suggested with details in tracked changes. I have added the word freestream as suggested. As explained above, all results involving P90 turbulence intensity have been removed to help shorten and focus the paper. I have clarified the source of IEC mean turbulence intensity as requested.

Page 12: Figure removed as suggested.

Pages 15-20: I now only retain stable and neutral relative shear classes to illustrate the two sets of ENOW coefficients. I have also removed all of Appendix B.

---

## Author Response (AR2)

**Further Revisions to Converging Profile Relationships for Offshore Wind Speed and Turbulence Intensity**
Gus Jeans
July 2024

**Response to Report #1 from Anonymous Referee #2**

No response required, but again I thank Remi for his engagement in the online discussion and subsequent constructive formal review comments.

**Response to Report #2 from Anonymous Referee #3**

Many thanks for reviewing my revised paper, I am very pleased to see recognition as a highly valuable contribution to convergence of IEC and ISO standards.  I am surprised this reviewer finds issues with the structure, because I was careful to ensure it already followed journal requirements [1].  However, I am pleased to revise the paper within a more traditional scientific paper structure, with the hope of making the work more accessible to a wider audience.  I again thank this referee and the handling editor for this opportunity to further improve the manuscript.

The journal does not specify abstract length, but I have followed this reviewer suggestion and prefer the more concise version.  The short summary was temporary, so ENOW participants could review everything in one document.

I have restructured the introduction as suggested.  I retain the equations in the second part, as I do consider them important for setting the stage and they also form a framework to the literature review.  I did consider moving what is now Section 1.2 into a Methodology Section 2, but it prepares readers for the structure of the paper, which is part of the introduction according to some reputable journals [2].

The new Data and Methods section is renamed and restructured as suggested, with some aspects related to methodology retained in the introduction as explained above.  However, Section 2 now starts with a cross reference to Section 1.2, to clarify the methodology connection.

Results and discussion sections are still combined, which is common practice [2].  It is easy to quickly find recent WES publications that follow a similar structure.

I have moved what was in a long conclusions section into a discussion summary then added a new more focussed conclusions section, with an emphasis on suggested future research.

**Other Changes**

I made a few other improvements that can be seen in the tracked changes, including new acknowledgements to the anonymous reviewers and relevant editors.
* * *
[1] https://www.wind-energy-science.net/submission.html#manuscriptcomposition

[2] https://www.nature.com/scitable/topicpage/scientific-papers-13815490/